# Pesticide Use and Serum Acetylcholinesterase Levels among Flower Farm Workers in Ethiopia—A Cross-Sectional Study

**DOI:** 10.3390/ijerph17030964

**Published:** 2020-02-04

**Authors:** Meaza Gezu Shentema, Abera Kumie, Magne Bråtveit, Wakgari Deressa, Aiwerasia Vera Ngowi, Bente E. Moen

**Affiliations:** 1Department of Preventive Medicine, School of Public Health, Addis Ababa University, Addis Ababa P.O. Box 9086, Ethiopia; aberakumie2@yahoo.com (A.K.); deressaw@gmail.com (W.D.); 2Department of Global Public Health and Primary Care, University of Bergen, 5020 Bergen, Norway; Magne.Bratveit@uib.no (M.B.); Bente.Moen@uib.no (B.E.M.); 3School of Public Health and Social Sciences, Department of Environmental and Occupational Health, Muhimbili University of Health and Allied Sciences, Dares Salaam 65015, Tanzania; vera.ngowi@gmail.com; 4Centre for International Health, University of Bergen, 5009 Bergen, Norway

**Keywords:** cholinesterase level, Ethiopia, flower farm workers, pesticides, WHO hazard classification

## Abstract

The flower industry in East Africa has grown in recent years, especially in the production and export of roses. The aim of this study was to assess pesticide use on selected flower farms in Ethiopia. Serum cholinesterase levels in workers were used as a marker of pesticide exposure. This study was a cross-sectional study involving 588 workers from 15 different flower farms. It had a response rate of 95.5%. The participants included 277 males (mean age 26 years; 148 pesticide sprayers and 129 non-sprayers) and 311 females (mean age 25 years; 156 working in greenhouses and 155 working outside the greenhouses). The researchers undertook structured interviews, blood sampling, and walkthrough surveys. Descriptive statistics and Poisson regression were used in the statistical analyses. A total of 154 different trade names of pesticides were found. Of them, 31 (27%) were classified as moderately hazardous by the WHO, and 9% were organophosphates. Serum levels of cholinesterase deviating from 50–140 Michel units were considered abnormal. Abnormal serum cholinesterase levels (above 140 Michel units) were found in 97 participants (16.5%, 95% confidence interval 13.7–19.7%). There were no differences between the four job groups regarding cholinesterase levels. The high prevalence of abnormal serum cholinesterase levels might indicate the presence of pesticide intoxication. Thus, there is a need for routine monitoring of all workers exposed to pesticides, not only sprayers.

## 1. Introduction

Pesticide use in agricultural sectors in Ethiopia has increased in recent years due to the rapid growth of flower farms [1,2,3]. The import of pesticides in 2002–2006 was 13,381 tons, and this increased to 30,059 tons from 2006–2011 [4]. This increase was likely caused by the large and growing demand related to agricultural crop production and flower farms [5]. The Government of Ethiopia designed a pesticide risk-reduction programme that ran from 2010 to 2015 and mainly focused on the sustainable management of pesticides in order to protect the health of workers and people living in the surrounding communities. There are also environmental concerns [6]. There are concerns that this increased pesticide use might enhance the risk of pesticide exposure among the workers and might cause serious adverse health effects in the farm worker community [5,7,8].

Pesticides are chemicals produced to control pests and thus improve agricultural productivity. However, pesticides are very toxic, and may have undesirable health effects in humans and other organisms [9]. The negative health effects of pesticides vary from short-term (skin irritation, eye irritation, headache, dizziness and nausea) to chronic health impacts (asthma, diabetes and cancer) [10]. Numerous pesticides are used on flower farms and represent different chemical and hazard classes, some of them belonging to the hazard class I and II according to the classification of the World Health Organization (WHO) [11,12,13]. According to the Ethiopian Ministry of Agriculture, around 300 chemicals are used by the flower industry in Ethiopia [14]. 

Workers in flower farms are mainly engaged in three different types of tasks. The first is planting, weeding, cultivating and harvesting flowers inside the greenhouses. This work is mainly performed by females. The second is spraying of pesticides inside the greenhouses, a task performed mostly by males. The third is packing the flowers in packinghouses outside the greenhouses, and both male and female workers are engaged in this work. Although many of the workers are not actively engaged in pesticide spraying activities, they can still be exposed to pesticides. Greenhouses are confined enclosures that allow the regulation of temperature and humidity to provide optimum growth conditions for flowers. Thus, workers working inside these greenhouses might be exposed to pesticides while they perform their daily activities due to pesticide aerosols that can be suspended in the air after the spraying process or due to skin contact with flowers treated by pesticides. Packinghouse workers can also be exposed to pesticides while handling and working with freshly harvested flowers [13,15]. The worker’s lack of personal protective equipment will worsen their exposure to the pesticides [5].

Information about pesticide exposure among the different workers on flower farms is very scarce. In addition, studying exposure to pesticides in this working sector is a challenge because of the number of different pesticides used. Flowers are ornamentals, and thus, authorities do not control the amount of pesticide residue on them. The exposure route can be both by inhalation and skin absorption [16]. In other settings, researchers use biomonitoring techniques involving urine and blood samples as the primary methods for assessing human exposure to pesticides [17].

Organophosphates and carbamates are commonly used pesticides on flower farms [1,3,5,13]. In living organisms, these pesticides are acetylcholinesterase inhibitors, meaning they inhibit the enzyme that normally breaks down the acetylcholine and stops the cholinergic activity of the neurons. This effect leads to a reduction in the acetylcholinesterase enzyme and a higher level of acetylcholine in serum. The measurement of the cholinesterase level in the blood of workers is a good indicator of human exposure to cholinesterase inhibitors [18] and is thus a good indicator of this type of pesticide exposure on flower farms. Serum cholinesterase levels (SChE) show a dose-relationship with organophosphate exposure. They are easy to measure, sensitive methods are available and the serum levels are also linked with adverse health effects in the exposed persons [19]. The method has been used in previous studies from Spain. For instance, in a study conducted by Hernandez et al., fifty percent of the workers exposed to organophosphate pesticides showed a 15% reduction in acetylcholinesterase level compared to baseline level [20]. In another study conducted in Spain, it was found that during the spraying season, about 64% of greenhouse workers had 25% lower SChE levels compared to what they had in the non-spraying season [21].

Previous studies from Ethiopia have shown the use of different pesticides on flower farms, including cholinesterase inhibitors. However, to our knowledge, no study has assessed pesticide exposure in flower farm workers using biomonitoring techniques. In addition, there have been no such studies in other parts of East Africa. Studies on cholinesterase levels conducted in other countries have focused on the working conditions related to pesticide spraying [3], but in the Ethiopian flower farms, there are many female workers engaged in cultivating, harvesting and post-harvesting activities, and who might be exposed to pesticides even though they are not directly involved in the pesticide spraying activity. Thus, the aim of this study was to assess pesticide use in flower farms and SChE levels among all flower farm workers in Ethiopia.

## 2. Materials and Methods

### 2.1. Study Area and Design

The study was conducted using a cross-sectional study design among workers in flower farms located within a 50-km radius of Addis Ababa, Ethiopia. In this area, there were 31 rose-producing farms. The area was selected because of the concentration of a large number of flower farms. Also, the need to bring blood samples quickly to the laboratory in Addis Ababa reduced the possibility of studying workers farther away from the city.

### 2.2. Sample Size Determination

The sample size was determined using a mean difference sample size calculation formula based on information from a study conducted among Nepalese mango farmers [11]. The calculation was made to find the number of workers needed to show differences in acetylcholinesterase levels between groups. We expected the pesticide sprayers in our study to be highly exposed to pesticides, and we wanted to compare them with workers not spraying on the farms. Because males and females might have different levels of cholinesterase activity, we separated males and females in the analysis [22]. We also expected that females working within greenhouses would be more exposed to pesticides than females working in packinghouses, and we wanted to compare these two groups as well. We calculated the number of workers needed using the mean acetylcholinesterase level for pesticide-exposed workers (sprayers) in the Nepali study of 28.92 Elman units/g with a standard deviation of 3.09, and the mean acetylcholinesterase level for controls of 30.05 Elman units/g, with a standard deviation of 2.66. The calculated sample size was 137 for each group when the statistical power was 90. We expected a 15% non-response rate, but the figure increased by 15%, and the final sample size calculated for each group was 158 workers (female greenhouse workers, female and male packinghouse workers, and male pesticide sprayers). For practical reasons, we decided to invite 160 participants from each group.

### 2.3. Selection of Farms and Sampling Procedure

Eight flower farms out of the 31 farms in the study area were randomly selected to participate in the study. The administrators of the selected farms were approached to obtain permission for the study after informing them about the procedures and purpose. They were also given time to discuss the issue with their respective farm owners and managers for a decision. Six farms agreed to participate, and two refused without giving any reason.

The number of female workers selected from each farm was in proportion to the size of the six farms in order to obtain the required total number of 160 workers from greenhouses and 160 workers from packinghouses. The total number of female greenhouse workers per farm ranged from 112 to 256, and the total number of packinghouse workers per farm ranged from 50 to 63 (Table 1). The total response rate was 97.5% for female greenhouse workers and 97% for the female packinghouse workers.

Male workers on the flower farms were few in number, ranging from 17 to 46, so we decided to include all male workers from all six farms. In addition, we randomly selected 10 additional farms for the study on male workers, of which nine agreed to participate. All sprayers and male packinghouse workers from these nine additional farms were invited to participate in the study. In total, 155 male sprayers and 141 male packinghouse workers were invited to participate, and 148 sprayers and 129 non-sprayers participated in the study, with an overall response rate of 93.6%. The administrators provided lists of workers to the researchers. The researchers selected the female workers using a systematic random sampling method, choosing every eighth greenhouse worker and every second packinghouse worker on the list. 

### 2.4. Interviews, Blood Sampling and Laboratory Analysis of Serum Cholinesterase Level

The workers were interviewed in a separate room on the farm where the privacy of the workers could be ensured. The principal investigator and a well-experienced research assistant who was trained for this purpose collected the data through face-to-face interviews using a structured questionnaire. The questionnaire asked about age, sex, educational status (categorized as illiterate, able to read and write, completed grades 1–4, completed grades 5–8, completed grades 9–10, completed grades 11–12, college and above), as well as ever drinking alcohol (yes/no) and ever smoking (yes/no).

In addition, the questionnaire included questions about working section (spraying, greenhouse, packinghouse), number of months working on the farm, transfer from another working section to the present (yes/no), number of working hours per day, previous work on other flower farms (yes/no), number of years working on other flower farms, working another job spraying pesticides (yes/no), whether they had their own farm (yes/no), and, if yes, if they had sprayed pesticides on their own farm in the last three months (yes/no). After the interview, the height and weight of the participants were measured and the body mass index (BMI) was calculated. 

After the interview and at the same location, a blood sample was collected from each study participant by a trained laboratory technician using a vacuum tube serum separator. Blood samples were collected on random days without consideration to whether spraying was done or not and without consideration of the types of pesticides sprayed during the data collection periods. In addition, the sampling we did was on an average workday and the researchers chose the days for sampling. The collected blood sample was immediately placed inside an ice pack, and the specimen was transported to the Food, Medicine, and Health Care Administration and Control Authority (FMHACCA) laboratory on the same day of its collection. At the laboratory, serum was separated by centrifugation and serum was stored in a refrigerator at the laboratory at a temperature of 2–8 °C until the analysis was complete. At the end of the data collection, all 588 samples were analysed for SChE using the electrometric “Michel” change-in-pH method at a rate of 40–60 samples per day [23]. 

The analysis of SChE was performed based on the standard operating procedures of the FMHACCA laboratory in Addis Ababa [23,24]. A buffer solution was prepared by dissolving 1.24 g of sodium barbital (0.006M), 0.14 g of KH_2_PO_4_ (0.001M) and 17.54 g of NaCl in about 900 mL of distilled water. Then, 11.6 mL of 0.1 M HCl was added, and finally, the solution was diluted to 1 L and mixed. A few drops of toluene were then added, and the solution was kept in a refrigerator until used.

To each 0.1 mL of serum sample, 5 mL of water and 5 mL of buffer solution was added, mixed well and incubated at 25 °C for 10 min, and pH_1_ was read and recorded after 10 min. After measuring pH_1_, 1 mL of 3% acetylcholine was added and allowed to incubate for 1 h at 25 °C, and pH_2_ was read and recorded. The above procedure was repeated without the addition of serum as a blank control. Michel units were calculated as [((pH_1_ − pH_2_) − (blank pH_1_ − blank pH_2_)) × 100]/(t_2_ − t_1_), where the time difference t_2_ − t_1_ is in this case was 1 h. 

Michel units in the range of 50–140 were considered to be normal values, and any result above or below this range was considered an abnormal value [23]. FMHACCA uses this cut of value to monitor workers’ exposure to cholinesterase inhibiting pesticides and determines workers who need to be transferred from high pesticide exposure working sections.

### 2.5. Collection of Data on the Types and Usage of Pesticides

#### 2.5.1. Walkthrough Survey

A walkthrough survey was performed in six of the farms using a checklist. The survey focused looking at the production flow, including cultivation of roses in the greenhouse, spraying of pesticides and packing of roses in the packinghouse. The principal investigator conducted the walkthrough survey with the help of production managers on each farm, who also answered some questions. The walkthrough was conducted in the same period in which the workers were examined in the present study.

The checklist included items related to the following: -The work process of pesticide mixing and spraying, greenhouse work and packinghouse work.-Pesticide mixing and storage. Chemical stores were visited and lists of pesticides in use with their active ingredients were obtained from storekeepers with permission from the farm managers.-The use of personal protective equipment by workers (boots, overalls, gloves, respirators, goggles and aprons).-Ventilation: Mechanical or natural (presence of windows).-Re-entry times in sprayed greenhouses. Production managers who assisted us in the walkthrough survey were asked how much time is required before workers can re-enter the greenhouse.

#### 2.5.2. Evaluation of Lists of Pesticides

The Ethiopian Ministry of Agriculture is authorised to have a list of registered pesticides in the country [22], and this list was obtained. According to Ethiopian pesticide registration and control proclamation No. 674/2010, “*No pesticides shall be registered unless the efficacy, safety and quality is tested under field or laboratory conditions and approved by the ministry. No person may formulate, manufacture, import, pack, repack, label, sell, distribute, store or use a pesticide not registered by the ministry*” [25]. The pesticide registration includes determining the safety and quality of pesticides [26,27]. Lists of pesticides used on the farms were obtained from the farms and checked against the Ministry’s lists to determine if the pesticides in use were registered in Ethiopia. In addition, we made a classification of pesticides based on their acute hazard and chemical type using the WHO-recommended pesticide classification and the Pesticide Action Network database of North America [28,29]. 

### 2.6. Data Management and Statistical Analysis 

The data were entered into EpiData version 3.1 software and exported to STATA version 14 (StataCorp, College Park, TX, USA) for further analysis. Descriptive statistics were calculated, including frequencies, percentages, means and medians. The study groups were compared using independent *t*-tests for continuous variables, and Pearson’s chi-squared test was used for comparing categorical variables.

Poisson regression with robust estimate of variance was computed to analyse the relationship between abnormal SChE and working in the different groups, analysing males and females separately. In the Poisson regression analysis model, we adjusted the data for age, BMI, education status, ever drinking alcohol and duration of service in months. The Poisson regression model has become a preferred approach than logistic regression in several cross-sectional studies with binary outcomes to produce the prevalence ratio (PR) estimates than the odds ratio (OR) in quantifying the relationship between independent variables and the outcome of interest [30,31,32]. The regression results are presented as PR with 95% confidence intervals (CIs) for the relationship between the independent variables and abnormal SChE. For all analyses, a value of *p <* 0.05 was taken to indicate statistical significance.

### 2.7. Ethical Consideration

Ethical approval was obtained from the Institutional Review Board of the College of Health Sciences at Addis Ababa University. Permission to conduct the study was obtained from each farm. Respondents were asked for written informed consent after being informed about confidentiality. The participation was voluntary, and participants were informed about the possibility of withdrawal at any time during the study. They were informed that the results of the study would be communicated back to the respective farms with aggregated numbers, with no personal identification of workers.

## 3. Results

### 3.1. Characteristics of the Participants 

In this study, 588 flower farm workers participated, with an overall response rate of 95.5%. Of these, 311 participants were female, including 156 greenhouse and 155 packinghouse workers, and the 277 male flower farm workers included 148 sprayers and 129 non-sprayers. The mean age was 25 years for females and 26 years for males, and there was no difference in age between the two female groups or between the two male groups. The mean numbers of months working at the current flower farm was 32, and there was no difference in the number of working months when comparing male and female workers or when comparing the two male and female study groups (Table 2). The mean working hours per day for the study groups was 7.3 h. Male sprayers worked fewer hours compared to male non-spraying workers, and female greenhouse workers reported working fewer hours compared to female packinghouse workers.

Female greenhouse workers had lower educational status, and 55 (17.6%) of the female workers and 33 (11.9%) of the male workers were unable to read and write. Twenty-three male and two female workers were ever smokers, while only six male workers were current smokers. One hundred and thirty-three workers (22.6%) had previously worked on other flower farms, of which 46 females and 13 males had worked in greenhouses, 18 males and 14 females had worked in packinghouses, 30 males had sprayed, 1 male had worked on pesticide mixing, and 17 males and 4 females had worked in other activities such as maintenance, cleaning or transportation.

### 3.2. Serum Cholinesterase Level

The overall prevalence of abnormal SCHE (above 140 MU) was 98 (16.7%, with a 95% CI 13.8–19.9%) (Table 3). The prevalence of abnormal SChE among female workers was 52 (16.7%) with a 95% CI (13.0–21.39%), and among males, it was 46 (16.6%) with a 95% CI 12.7–21.5%). 

There was no significant difference in abnormal SChE level when comparing female greenhouse workers and female packinghouse workers, no difference between male sprayers and male packinghouse workers and no difference between spraying and overall non-spraying workers. The prevalence ratio also showed no differences between the groups with total working years below or above five years (Table 3).

### 3.3. Pesticide Use in the Flower Farms

#### 3.3.1. Workplace Assessment

Walkthrough surveys were conducted on six farms, and the activities of the four groups of workers and their contact with pesticides were observed. Pesticides were usually mixed in a separate room designated for only mixing processes, but some used the same room for pesticide mixing and storage. Mixing rooms in many of the observed farms were confined rooms, and on two of the farms, there were no windows for ventilation in the mixing room. There were usually one or two pesticide mixers per farm, and after mixing the pesticides, the substances were pumped to the greenhouses through a pipe attached to the container. Protective equipment used by the pesticide mixing workers included half facemasks (respirator masks), gloves, boots and overalls. On many of the farms, the overalls were made of textile and were not chemical-proof.

The sprayers in the greenhouses were all males. In each greenhouse, there were pipes connected to the mixing container from the mixing rooms, and the sprayers connected the pipe to the spraying wand and sprayed the pesticides onto the flowers manually. The workers in all the observed farms used a forward-spraying technique whereby the worker spraying walked toward the spraying cloud. The personal protective equipment included half facemasks, gloves and overalls. The masks used by these workers were labelled with either a brown or yellow colour indicating protection against organic gases and vapours and against SO_2_ and other acid gases, respectively. The sprayers usually used respirators while they were spraying, but when they were waiting for their turn to spray, most of them took off their respirators even though they were still inside the greenhouse being sprayed. The gloves used by sprayers were long-sleeved rubber gloves. The overalls on three of the farms were simple textile, and on two farms, the overalls were plastic. Aprons and goggles were not used by any of the workers observed during spraying activities. The spraying activity was only observed on five of the six farms because, on one farm, spraying took place during the night-time, which was inaccessible for observers. 

Females worked in greenhouses where they cultivated flowers. The tasks performed by female workers included weeding, examining flowers for disease, harvesting flowers and transferring flowers to the packinghouse. The greenhouse workers in many of the observed farms wore gloves and aprons, and textile overalls were used on a few farms. The gloves used by the greenhouse workers were either leather or fabric, and on many of the farms, the workers wore perforated gloves. On many of the farms, workers were evacuated to other greenhouses when spraying was going on, and there were rules regarding when they could re-enter the sprayed greenhouses. However, on two farms, workers were observed working in the same greenhouse section while spraying took place. The re-entry time was reported to be different according to the type of pesticide and ranged from 2 h to nearly half a day. In many of the cases, these workers wore simple clothes, such as skirts or dresses, that might increase their chance of exposure through the bare skin on their legs coming into contact with treated flowers. 

The harvested flowers brought into the packinghouse might have been recently sprayed. Packinghouse workers were engaged in arranging and packing flowers to be exported, and both females and males were involved in these activities. The male workers mainly trimmed leaves using scissors and removed loose flower petals by beating the flowers. They worked in a standing position and the leaves and petals fell onto the floor. In addition, these workers were involved in transporting flowers from the main packinghouse to the cold storage room. These workers used protective equipment such as overalls and gloves, but in the middle of their work, some workers removed their gloves. Female packinghouse workers were mainly involved in cutting flower stems to the required size, making bundles and packing bundles of flowers. They wore gloves made of simple fabric to protect themselves from thorns. However, workers were sometimes observed with gloves on only one hand. The males working in this section primarily wore cloth overalls in addition to gloves. Within the packinghouse, there was also a cold room with temperatures below 3 °C where flowers were stored. 

#### 3.3.2. Types of Pesticides Used on the Flower Farms

A total of 154 different trade names of pesticides were identified from all farms, ranging from 22 to 50 pesticides on each farm. Only 45 (29.2%) of these pesticides were registered by the Ethiopian Ministry of Agriculture. Among the 154 total pesticides listed, the active ingredients of 113 could be accessed. These were classified based on their chemical type and the WHO acute hazard classification (Table 4). Only one pesticide, an organophosphate, was classified as class IB, a highly hazardous active ingredient, while 31 (27.4%) of the pesticides were in class II, or moderately hazardous. Pesticides with different chemical compositions were used on the farms, including organophosphates, pyrethroids, neonicotinoids, inorganic pesticides, azoles, dicarboximides, morpholine, dithiocarbamate, pyrazole, pyrimidines, benzoylurea, silicone and anilide, as well asmicrobial and botanical agents. Organophosphates, neonicotinoids and pyrethroids made up 30 (26.7%) of the total pesticides reported. Ten (8.9%) of the pesticides used were organophosphates, and all farms reported the use of at least one organophosphate pesticide. The highest number of organophosphates in use was six, which was found on one farm (Table 4).

## 4. Discussion

The WHO classifies the majority of the pesticides used on the surveyed farms as moderately hazardous. Organophosphate pesticides were used on all surveyed farms. SChE levels showed abnormally high values among 16.5% of the flower farm workers, but there was no significant difference in abnormal cholinesterase levels across the different working sections. 

In our study, an active ingredient was identified in 113 of 154 pesticides, which is comparable to a study conducted in the Netherlands which found 116 active ingredients used by greenhouse flower farms [16]. In addition, the majority of pesticides used by the studied flower farms were not registered by the Ministry of Agriculture, which is authorised to regulate pesticides within the country. Despite the strong set of regulations regarding the need for pesticide registration and to only use registered pesticides, pesticides which were not registered by Ethiopian ministry of agriculture were reported among the studied flower farms. Studies in Tanzania have also shown the use of unregistered pesticides by farm workers in general and on horticulture farms in particular [12]. The use of unregistered pesticides has also been reported in greenhouse farms in Turkey [33]. This might be attributed to the direct import of pesticides by commercial flower farms and inadequate pesticide controls, which was also reported by Negatu et al. [5].

The use of WHO class II (moderately hazardous) pesticides was also reported in previous studies in Ethiopia [5,13]. In our study, we also found the use of different classes of pesticides, including organophosphates, neonicotinoids and pyrethroids, which made up about 30% of the total number of pesticides found in this study. Organophosphates were used by all surveyed farms, which is in line with previous studies in flower farms in different countries [3,5,12,13]. Thus, the presence of organophosphates in all the surveyed farms confirms the possibility of workers being exposed to cholinesterase inhibitors.

The prevalence of abnormal cholinesterase levels in workers in the present study was lower than those found in a study among cut flower farm workers in Trinidad, where depressed levels of cholinesterase were found in the red blood cells of 25% of the workers. This difference in results might be explained by the different methods used for analysis of cholinesterase levels, as they used red blood cell cholinesterase level while we used SChE levels. Abnormal SChE was also more prevalent (32%) among Thai chilli workers compared to the workers in the present study [34]. Even though these groups of workers are not comparable to flower farm workers, both study groups used organophosphate pesticides. The differences in their cholinesterase levels might be due to different methods used to measure cholinesterase. In our study, we used the Michel change-in-pH method, while the study on Thai workers used the spectrometer-based Elman technique. Also, the working conditions of the Thai farmers might also be worse than in Ethiopia because the Ethiopian farms in our study are institutionalised and might be better at enforcing safety behaviours such as not eating, smoking or drinking in sprayed areas and wearing better personal protective equipment, at least while spraying.

The prevalence of abnormal SChE levels was higher compared to two other studies conducted in Thailand, which were 12% and 8.5% [35,36]. The discrepancy between these studies might be because of the sample size differences and agricultural type. The studies conducted in Thailand involved general agricultural workers, and these workers might be less exposed than those who are involved in intensive greenhouse farming. Abnormal cholinesterase level is an indication of worker exposure to pesticides. Thus, it is important to regularly survey these levels to monitor the worker health on flower farms to reduce the potential hazard effects [37,38].

There was no statistical difference in the prevalence of abnormal SChE levels when comparing different groups in the present study. One might have expected the male sprayers to be more exposed to pesticides than the other workers. However, the present findings indicate that all workers, not only the sprayers, were exposed to the pesticides, or at least to the acetylcholinesterase inhibitors. Workers in the greenhouses might be exposed to high levels of pesticides because they enter the greenhouses after spraying, and also in some of the cases, workers were observed inside the same time that the spraying was taking place. In a study conducted on large-scale greenhouses and open field farms in Ethiopia, cumulative pesticide exposure was found to be higher among re-entry workers in large-scale greenhouses compared to re-entry workers in open field farms [6]. Re-entry workers in that study referred to workers who were not spraying pesticides but who were involved in fieldwork of cultivating, harvesting, and packing agricultural products [6]. This indicates that exposure to pesticides can occur among workers on greenhouse farms even though they are not directly involved in spraying. This could also happen due to the presence of pesticide residues on flowers while they are being cultivated, harvested and further processed for export [16,39]. In addition, although pesticide inhalation is a concern in greenhouse farms because pesticides could remain in the air and workers could inhale pesticide mists, the majority of occupational exposure is reported to occur through the dermal route. It has been estimated that 58% of pesticide exposure occurs through dermal absorption by workers who are in contact with flowers during their entire work shift [40]. Thus, workers who are engaged in cultivating flowers inside greenhouses and who work in bundling and packing flowers might be exposed through the handling of harvested flowers. Exposure studies are needed to confirm these suggestions. Another reason for the presence of similar abnormal SChE in spraying and non-spraying workers might be due to environmental pesticide exposure in their residences due to the drift of pesticides from both open field and greenhouse farms. We have no information about the overall environmental exposure, but in support of this argument, a study conducted in Chile and Thailand found no significant difference in cholinesterase levels among study participants who were occupationally exposed and those who were environmentally exposed [41,42].

One of the strengths of this study is the very high response rate. Another is that we were able to select the study participants without any interference from administrators of the farms, thus increasing the validity of the study. We used the Michel method, which is a validated method for measuring cholinesterase levels, and the laboratory we used was a nationally accredited laboratory that regularly measures cholinesterase levels among exposed farm workers. However, relatively few published studies have used this method, and we therefore had few studies with which to compare our results. The inclusion of workers of different categories was another important strength of this study. However, better information about the pesticide exposure would have been very useful. However, due to the high number of pesticides used in the farms, this was difficult to achieve. Also, it would have been of interest to study the relevant health effects of the workers, not only the serum cholinesterase levels. This will be considered for future studies. One of the limitations of this study is the failure to include a control group of participants outside the flower farms and without pesticide exposure. Also, we would have benefited from repeated measurements of cholinesterase levels. Determining baseline level of cholinesterase for each individual would also have been important because of the wide individual range for cholinesterase [37,38]. We were not able to perform a study including the worker’s baseline serum cholinesterase levels because the workers were already working for a while at the time of the study. The follow-up of newly employed workers would have been of interest, but logistically very difficult to perform. The workers were also not a stable population group, as seen by the short working time for most of them, and the fact that the workers migrate from area to area makes the logistics of follow-up studies difficult.

Another limitation of the study is the failure to record the amount of pesticides used, the frequencies of spraying, and the re-entry time. Also, some farms did not want to participate in our study, and we have no knowledge of the exposure or health situation on these farms. 

## 5. Conclusions and Recommendations

The prevalence of abnormal SChE level (above 140 MU) among 16.5% of the workers is a finding that needs further attention and follow-up. It is likely that these levels are due to pesticide exposure in the farms. Large numbers of other pesticides were also used on the studied flower farms, and many of them were moderately hazardous according to the WHO classification. Many of the pesticides used were not officially registered. Further studies need to be conducted focusing on the exposure assessments and health effects of various pesticides used in flower farms. The consideration of protective and preventive measures seems to be needed to reduce the cholinesterase levels among the workers.

## Figures and Tables

**Table 1 ijerph-17-00964-t001:** Numbers of invited and participating female workers from six randomly selected flower farms in Ethiopia, 2017.

Workers	Total	Farm					
		1	2	3	4	5	6
Total number of female greenhouse workers	1282	225	112	256	242	237	210
Total number of invited female greenhouse workers	160	28	14	32	30	30	26
Number of participating female greenhouse workers	156	27	13	32	30	30	24
Total number of female packinghouse workers	324	54	50	60	63	57	40
Total number of invited female packinghouse workers	160	28	25	29	30	28	20
Number of participating female packinghouse workers	155	28	23	28	30	26	20

**Table 2 ijerph-17-00964-t002:** Characteristics of participating flower farm workers in a study of pesticide exposure in Ethiopia, 2017.

Variables		All Workers	Working Section Females/Males						Sex		
			Green-House (*n* = 156)	Packinghouse (*n* = 155)	*p*-Value	Sprayers (*n* = 148)	Non-Sprayers (*n* = 129)	*p*-Value	Male (*n*= 277)	Female (*n* = 311)	*p*-Value
Mean age in years (SD)		25 (7)	25 (8)	24 (7)	0.08 ^1^	27 (6)	26 (7)	0.66 ^1^	26 (7)	25 (7)	<0.01 ^1^
Mean working months on current farm (SD)		32 (33)	30 (31)	31 (27)	0.66 ^1^	32 (31)	38 (43)	0.17 ^1^	34 (37)	31 (29)	0.16 ^1^
Mean working hours (SD) per day		7.4 (1.3)	7.9 (0.5)	8 (0.3)	0.03 ^1^	5.7 (1.6)	7.9 (0.5)	<0.01 ^1^	6.7 (1.6)	7.9 (0.4)	<0.01 ^1^
Mean body mass index (SD)		20.3 (2.4)	20.6 (2.6)	20.8 (2.5)	0.74 ^1^	19.9 (2.0)	19.8 (2.1)	0.47 ^1^	19.8 (2.0)	20.7 (2.6)	<0.01 ^1^
Educational level	Unable to read and write *n* (%)	88 (20.0)	39 (25.0)	16 (10.3)	<0.01 ^2^	19 (12.8	14	0.61 ^2^	33 (11.9)	55 (17.7)	0.05 ^2^
	Able to read and write *n* (%)	500 (80.0)	117 (75.0)	139 (89.7)		129	115		244 (88.1)	256 (82.3)	
Transferred ever from other work section	Yes *n* (%)	62 (10.5)	2 (1.3)	22 (14.2)	<0.01 ^2^	17 (11.5)	21 (16.3)	0.251 ^2^	38 (13.7)	24 (7.7)	0.02 ^2^
	No *n* (%)	526 (89.5)	154 (98.7)	133 (85.8)		131 (88.5)	108 (83.7)		239 (86.3)	287 (92.3)	
Worked on other flower farm	Yes *n* (%)	133 (22.6)	32 (20.5)	32 (20.7)	0.97 ^2^	36 (24.3)	33 (25.6)	0.80 ^2^	69 (24.9)	64 (20.6)	0.21 ^2^
	No *n* (%)	455 (77.4)	124 (79.5)	123 (79.4)		112 (75.7)	96 (74.4)		208 (75.1)	247 (79.4)	
Ever drinking alcohol	Yes *n* (%)	291 (49.5)	56 (35.9)	51 (32.9)	0.56 ^2^	86 (58.9)	98 (75.9)	<0.01 ^2^	184 (66.4)	107 (34.3)	<0.01 ^2^
	No *n* (%)	297 (50.5)	100 (64.1)	104 (67.1)		62 (41.9)	31 (24.0)		93 (33.6)	204 (65.6)	
Own farm	Yes *n* (%)	105 (17.9)	11 (7.1)	7 (4.6)	0.34 ^2^	58 (39.2)	29 (22.5)	<0.01^2^	87 (31.4)	18 (5.8)	<0.01 ^2^
	No *n* (%)	483 (82.1)	145 (93.0)	148 (95.5)		90 (60.8)	100 (77.5)		190 (68.6)	293 (94.2)	

SD = standard deviation; ^1^ = *t*-test; ^2^ = Pearson’s chi-square test.

**Table 3 ijerph-17-00964-t003:** Serum cholinesterase (SChE) level of workers with respect to different working sections among males, females and the total number of flower farm workers.

Characteristics		SCHE in MU AM(SD)	*p*-Value (*t*-Test)	Range	No. of SCHE above 140 MU (%) (95% CI)	Prevalence ratio 95% CI
Female workers (*n* = 311)						
Work section	Greenhouse workers (*n* =156)	116.2 (29.3)	0.42	51–189	27 (17.4) (12.1–24.1)	0.9 (0.89–1.09)
	Packinghouse workers (*n* = 155)	118.7 (25.3)		50–190	25 (16.1 (11.1–22.8))	1
Service year	<5 years (*n* = 244)	118.1 (27.8)	0.42	50–190	42 (17.2) (13.0–22.5)	1.04 (0.91–1.18)
	≥5years (*n* = 67)	115.1 (27.5)		69–189	10 (14.9) (8.2–25.7)	1
Transferred from Other work section	Yes (*n* = 24)	116.1 (26.0)	0.65	81–174	2 (8.3) (2.0–28.6)	1.11(0.94−1.30)
	No (*n* = 287)	117.8 (27.7)		50–190	50 (17.4) (13.4–22.3)	
Male workers (*n* = 277)						
Work section (*n* = 277)	Sprayers (*n* = 148)	116.1 (25.9)	0.85	77–190	23 (15.5) (10.5–22.4)	1.03 (0.29–1.14)
	Non-sprayers (*n* = 129)	115.5 (24.2)		64–164	23 (17.8) (12.1–25.5)	1
Working years	<5 years (*n* = 218)	115.3 (25.0)	0.49	70–190	36 (16.5) (12.1–22.1)	1.04 (0.89–1.21)
	≥5 years (*n* = 59)	117.8 (25.4)		64–189	10 (16.9) (9.3–28.9)	
Transferred from other work section	Yes (*n* = 38)	117.2 (22.8)	0.15	70–164	7 (18.3) (8.9–34.2)	0.99 (0.84–1.16)
	No (*n* = 239)	112.1 (25.6)		64–190	39 (16.3) (12.1–21.6)	
Own farm	Yes (*n* = 87)	118.6 (25.5)	0.21	76–183	13 (14.9) (8.8–24.2)	1.04 (0.92−1.17)
	No (*n* = 190)	114.6 (24.8		64–190	33 (17.4) (13.6–23.5)	
Total workers (*n* = 588)						
Work section	Sprayers (*n* = 148)	116.1 (25.9)	0.75	77–190	23 (15.3) (10.5–22.3)	1.03 (0.95–1.12)
	Non-sprayers (*n* = 440)	116.9 (26.4)		50–190	75 (17.05) (13.8–20.9)	1
Working years	<5 years (*n* = 462)	116.8 (26.5)	0.87	50–190	78 (16.9) (13.7–20.6)	1
	≥5 years (*n* = 126)	116.4 (25.4)		64–189	20 (15.9) (10.4–23.4)	1.01 (0.92–1.10)
Transferred from other work section	Yes (*n* = 62)	117.9 (22.9)	0.70	70–174	9 (14.5 (7.7–25.7))	1.03 (0.92–1.15
	No (*n* = 526)	116.626.7)		50–190	89 (16.9 (13.9–20.4))	1

AM = arithmetic mean, SD = standard deviation, SChE = serum cholinesterase level, MU= Michel units, CI = confidence interval.

**Table 4 ijerph-17-00964-t004:** Pesticide classification based on WHO acute hazard classification and the chemical composition of the pesticides’ active ingredients.

Classification		All Farms n = 113 No. (%)	Farm 1 n = 25 No. (%)	Farm 2 n = 16 No. (%)	Farm 3 n = 29 No. (%)	Farm 4 n = 36 No. (%)	Farm 5 n = 31 No. (%)	Farm 6 n = 50 No. (%)
WHO classification	Highly hazardous (IB)	1 (0.9)	1 (4.0)	0	0	0	0	0
	Moderately hazardous (II)	31 (27.4)	3 (12.0)	9 (56.3)	6 (20.7)	5 (13.9)	10 (32.3)	18 (36.0)
	Slightly hazardous (III)	17 (15.0)	3 (12.0)	3 (18.8)	7 (24.1)	7 (19.4)	4 (12.9)	4 (8.0)
	Unlikely to be hazardous	23 (20.4)	5 (20.0)	2 (12.5)	5 (17.2)	10 (27.8)	7 (22.5)	8 (16.0)
	Classes could not be traced	41 (36.3)	13 (52.0)	2 (12.5)	11 (37.9)	14 (38.9)	10 (32.3)	20 (40.0)
Chemical Classes of Pesticides	Organophosphates	10 (8.9)	2 (8.0)	2 (12.5)	5 (17.2)	1 (2.8)	3 (9.7)	6 (12.0)
	Neonicotinoids	10 (8.9)	1 (4.0)	3 (18.8)	2 (6.9)	4 (11.1)	3 (9.7)	3 (6.0)
	Pyrethroids	10 (8.9)	1 (4.0)	0	1 (3.4)	3 (8.3)	1 (3.2)	5 (10.0)
	Inorganic	9 (8.0)	2 (8.0)	1 (6.3)	4 (13.8)	0	1 (3.2)	4 (8.0)
	Unclassified	52 (46.0)	3 (12.0)	0	3 (3.4)	10 (27.8)	9 (29.0)	4 (8.0)
	Others	22 (19.5)	16 (64.0)	10 (62.5)	14 (48.3)	18 (50.0)	14 (45.2)	28 (56.0)

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
