# Peer review of "Pesticide Use and Serum Acetylcholinesterase Levels among Flower Farm Workers in Ethiopia—A Cross-Sectional Study"

_ijerph, 2020, doi:10.3390/ijerph17030964_

Round 1

Reviewer 1 Report

This paper describes a large cross-sectional survey of pesticide use and serum cholinesterase levels across different groups of workers on flower farms (greenhouse, spraying, packing) in East Africa. The descriptive findings on pesticides were informative, confirming use of a large number of products with diverse active ingredients, including non-registered pesticides and organophosphates. While the sample size was large, the study did not identify differences between groups. It may have been difficult to detect possible differences given gender-specific tasks and time worked under variable conditions and with variable protective equipment. Unfortunately, the design did not include an unexposed control group or pre- or post-work specimens. The lack of differences in serum cholinesterase levels is somewhat notable, however, suggesting that greenhouse workers with longer hours worked and different types of tasks and protections had similar levels of depression from chronic exposures compared with those exposed more intensely (but for shorter periods) due to spraying the pesticides. Comparisons to other populations is difficult due to the nature of the assay.

The authors could further discuss the ideal window for sampling and implications of "spot" measurements capturing a single point in time - is this an average workday. 

Was there any way to identify which farms used more organophosphates? Given the lack of overall differences between types of workers, perhaps there is room to compare levels between farms? 

The paper is clearly written, and the authors do a good job describing the design, the findings, and discussing the limitations of the study. The study design limitations cannot be helped at this point, but the results point to a need for a more focused investigation on symptoms and health effects, with a more rigorous design accounting for knowledge gained from this initial surveillance. Given the age of the population and high proportion of female workers, it would also be important to know the pregnancy status of the women, as this may impact metabolism of pesticides as well as the potential impacts of exposures in this susceptible developmental window. The authors may want to take this opportunity to further highlight the these implications and needs in worker health and safety. 

Reviewer 2 Report

Dear authors,

some suggestions to improve the paper.

1) In introduction yoou should explain the health effects of pesticides. I suggest to add this:

Effects of pesticides on health vary from short-term (e.g., skin and eye irritation, headaches, dizziness, and nausea) to chronic impacts (e.g., asthma, diabetes and cancer such as Monoclonal gammopathy of uncertain significance (MGUS) and others). The risk depends on the involvement of various factors (e.g., period and level of exposure, type of pesticide (regarding toxicity and persistence), and the environmental characteristics of the affected areas). (cite: 1) Kim KH, Kabir E, Jahan SA. Exposure to pesticides and the associated human health effects. Sci Total Environ. 2017 Jan 1;575:525-535. doi: 10.1016/j.scitotenv.2016.09.009. Epub 2016 Sep 7. 2) Taino G, Bordini L, Sarto C, Porro S, Chirico F, Oddone E, Imbriani M Monoclonal gammopathy of uncertain significance (MGUS) and occupational risk factors: criteria to carry out the health surveillance. G Ital Med Lav Ergon. 2019 Jul;41(3):202-207).

2) In abstract you should add something about implications of your research for policymakers.

3) In methods you should explain better why you did use the cut off level of 140.

4) In discussion you could add something about the importance of health surveillance at workplace for the prevention of this Hazard. 
